# Testing the climate intervention potential of ocean afforestation using the Great Atlantic *Sargassum* Belt

Lennart T. Bach [1✉], Veronica Tamsitt[2,3], Jim Gower[4], Catriona L. Hurd[1], John A. Raven[5,6,7] & Philip W. Boyd [1]

Ensuring that global warming remains <2 °C requires rapid $CO_2$ emissions reduction. Additionally, 100–900 gigatons $CO_2$ must be removed from the atmosphere by 2100 using a portfolio of $CO_2$ removal (CDR) methods. Ocean afforestation, CDR through basin-scale seaweed farming in the open ocean, is seen as a key component of the marine portfolio. Here, we analyse the CDR potential of recent re-occurring trans-basin belts of the floating seaweed *Sargassum* in the (sub)tropical North Atlantic as a natural analogue for ocean afforestation. We show that two biogeochemical feedbacks, nutrient reallocation and calcification by encrusting marine life, reduce the CDR efficacy of *Sargassum* by 20–100%. Atmospheric $CO_2$ influx into the surface seawater, after $CO_2$-fixation by *Sargassum*, takes 2.5–18 times longer than the $CO_2$-deficient seawater remains in contact with the atmosphere, potentially hindering CDR verification. Furthermore, we estimate that increased ocean albedo, due to floating *Sargassum*, could influence climate radiative forcing more than *Sargassum*-CDR. Our analysis shows that multifaceted Earth-system feedbacks determine the efficacy of ocean afforestation.

[1] Institute for Marine and Antarctic Studies, University of Tasmania, Hobart, TAS, Australia. [2] University of New South Wales, Sydney, Australia. [3] Centre for Southern Hemisphere Oceans Research, CSIRO Oceans and Atmosphere, Hobart, TAS, Australia. [4] Fisheries and Oceans Canada, North Saanich, BC, Canada. [5] Division of Plant Sciences, University of Dundee at the James Hutton Institute, Invergowrie, Dundee, UK. [6] Climate Change Cluster, University of Technology, Ultimo, Sydney, Australia. [7] School of Biological Sciences, University of Western Australia, Crawley, WA, Australia. ✉email: Lennart.bach@utas.edu.au

Capturing atmospheric $CO_2$ using ocean afforestation is receiving widespread attention in scientific and public discourse, due to the prospects of converting seemingly unproductive open ocean deserts into thriving seaweed ecosystems[1–5]. Indeed, high-level political and economic stakeholders are beginning to advocate the rapid implementation of offshore seaweed farming[6,7], even though little is known about the CDR potential and side-effects of ocean afforestation when upscaled. Model simulations are the first step to explore these unknowns[8], but they inevitably miss many complexities associated with real-world systems. Therefore, performing in situ experiments must follow to inform model developments. However, most of what is experimentally and logistically feasible is orders of magnitude lower than the scale envisioned for climate-relevant ocean afforestation[9].

Studying natural analogues presents a low-risk, cost-effective way to evaluate the potential of ocean afforestation on the scale needed to contribute significantly to CDR. Indeed, there are prominent examples where studying natural analogues of climate intervention, such as the Mount Pinatubo volcanic eruption[10] or glacial–interglacial ocean iron fertilisation[11], has led to breakthroughs in our understanding of the response of the Earth System to perturbation. The recent blooms of floating seaweed *Sargassum* in the (sub)tropical North Atlantic[12,13] provide the first natural analogue for seaweed farming distributed across multiple open ocean regions. While the extent of these patchy *Sargassum* blooms (up to 6100 km²) are still below the maximum extent envisioned by some for ocean afforestation (e.g., 9% of the surface ocean[4]), they are orders of magnitude larger than anything that is experimentally feasible. Thus, the recent emergence of this Great Atlantic *Sargassum* Belt (GASB)[13] offers the unique, perhaps the only, opportunity to study large-scale ocean afforestation under real-world conditions before its potential application.

Floating *Sargassum* seaweed was historically found in the Sargasso Sea, but extended its bloom formations from 2011 onward across the Intra-Americas Sea and the (sub)tropical North Atlantic from West Africa to South America[12,13] (Fig. 1a–c). This extension may have been initiated by an extreme negative phase of the North Atlantic Oscillation during 2009–2010 which shifted regional wind patterns[14]. The subsequent development of the GASB is hypothesized to be sustained by increased nutrient runoff from the Amazon River[12,13]. The *Sargassum* bloom, detected using satellites, follows a seasonal cycle with highest biomass in summer[13,15] (Fig. 1d). The largest GASB bloom was observed in 2018 covering up to 6100 km² distributed in numerous rafts over a ~9000 km belt just north of the Equator[13,16]. The 2018 bloom commenced in November 2017 and peaked in June 2018 with a net build-up of 0.81 million tons (Mt) of particulate organic carbon (POC; Fig. 1d and Supplementary Table 1). This constitutes a lower bound as satellites neither detect small *Sargassum* rafts[13] nor biomass submerged transiently by wind-driven ocean circulation, which could increase *Sargassum* stocks[17]. POC formation in the GASB could be viewed as a first-order estimate for its capacity to absorb and sequester $CO_2$. However, this CDR potential will be modified through biogeochemical feedbacks. Here, we utilize insights from the GASB to elucidate the complex influence of calcification and nutrient reallocation on ocean afforestation.

## Results and discussion

### Biogeochemical feedbacks reduce CDR by ocean afforestation.

POC formation via photosynthesis consumes $CO_2$. In contrast, the formation of particulate inorganic carbon (PIC) through calcification (e.g., $Ca^{2+} + CO_3^{2-} \rightarrow CaCO_3$) releases $CO_2$ by reducing seawater alkalinity[18,19]. Thus, to accurately assess the CDR potential of the GASB, we must subtract the $CO_2$ formed via calcification from the $CO_2$ consumed via photosynthesis. *Sargassum* and other fleshy seaweeds do not calcify themselves, but provide a habitat for colonising epibiont calcifiers such as bryozoans[20]. Carbonate biominerals attached to *Sargassum* collected from the Sargasso Sea contribute 9.4% on an annual average to its wet weight biomass[20], equivalent to a PIC:POC ratio of ~0.25 (mol:mol). This reduces the $CO_2$ removal generated through photosynthetic POC formation by ~17% (Fig. 2a, c). It is currently unclear if 9.4% is applicable for the new *Sargassum* blooms occurring since 2011 in the GASB, or for other seaweeds potentially used for ocean afforestation. Slower/faster growing seaweeds may provide more/less time for epibiontic calcifiers to settle, which would affect the PIC:POC ratio accordingly. Over the range of wet weight $CaCO_3$ percentages reported for individual *Sargassum* samples from the Sargasso Sea (i.e., 4.3–21.4%)[20], the PIC:POC is 0.11–0.9 and the offset to $CO_2$ removal 7–57% (Fig. 2c). This indicates that the calcification offset could range from being negligible to being a major factor reducing the CDR efficiency of ocean afforestation.

Another reduction in the CDR potential of the GASB is due to reallocation of nutrient resources from phytoplankton to *Sargassum*. Photosynthesis by *Sargassum* consumes nutrients which would otherwise fuel phytoplankton photosynthesis. Therefore, we must subtract the amount of natural phytoplankton CDR that would have been possible, with the amount of nutrients reallocated to *Sargassum* CDR. In the open (sub)tropical North Atlantic, Nitrogen (N) limits primary production while Phosphorus (P) and Iron (Fe) are occasionally co-limiting[21]. The 2018 GASB consumed 2.7, 0.12, and 0.003 Gmol of N, P, and Fe, respectively. With these resources, phytoplankton could photosynthesise 0.26 Mt POC (assuming N-limitation[21] and a C:N ratio of 8 mol:mol[22]), thereby reducing *Sargassum* CDR by 32% (31% reduction assuming P-limitation and a C:P ratio of 170 mol:mol[23]; Supplementary Fig. 1). A key question is whether phytoplankton POC formation and the associated natural CDR would be equally effective as CDR by *Sargassum*. Phytoplankton in the open (sub)tropical North Atlantic maintain surface nutrient concentrations close to zero, which is indicative of a ~100% efficiency of the phytoplankton carbon sequestration in this region[24,25]. Thus, arguably, all phytoplankton POC that would have been fixed with the nutrients utilised by *Sargassum* must be subtracted from the *Sargassum* CDR, although this discount may be mitigated by the use of external nutrient sources (see also Supplementary discussion 1).

Nutrient reallocation from phytoplankton to *Sargassum* not only reduces phytoplankton POC, but also its PIC formation (i.e., calcification) and therefore the associated $CO_2$ formation. Hence, less $CO_2$ is formed by phytoplankton calcification because of the nutrient reallocation to *Sargassum*, and this saved amount of $CO_2$ must be added to the CDR potential of *Sargassum*. However, this addition is minor because the planktonic PIC:POC is small in the (sub)tropical North Atlantic (~0.01)[26,27] relative to *Sargassum* assemblages (PIC:POC ~0.25; see above).

Considering the offsets through calcification and nutrient reallocation we define the theoretical CDR potential of ocean afforestation as:

$$CDR_{theoretical} = POC_{seaweed} - PIC_{seaweed} - POC_{plankton} + PIC_{plankton}, \quad (1)$$

with the associated amounts of $CO_2$ bound/released given in Mt C. With respect to the 2018 GASB, $POC_{seaweed}$, $PIC_{seaweed}$, $POC_{plankton}$, and $PIC_{plankton}$ are 0.81, 0.13, 0.26, and 0.002 Mt C, respectively. Accordingly, the 2018 GASB generated a $CDR_{theoretical}$ of ~0.42 Mt C due to the higher organic C:N ratio of *Sargassum* (C:N = ~25) than phytoplankton biomass (C:N = 8)[22], and because this C:N "advantage" is not fully offset by

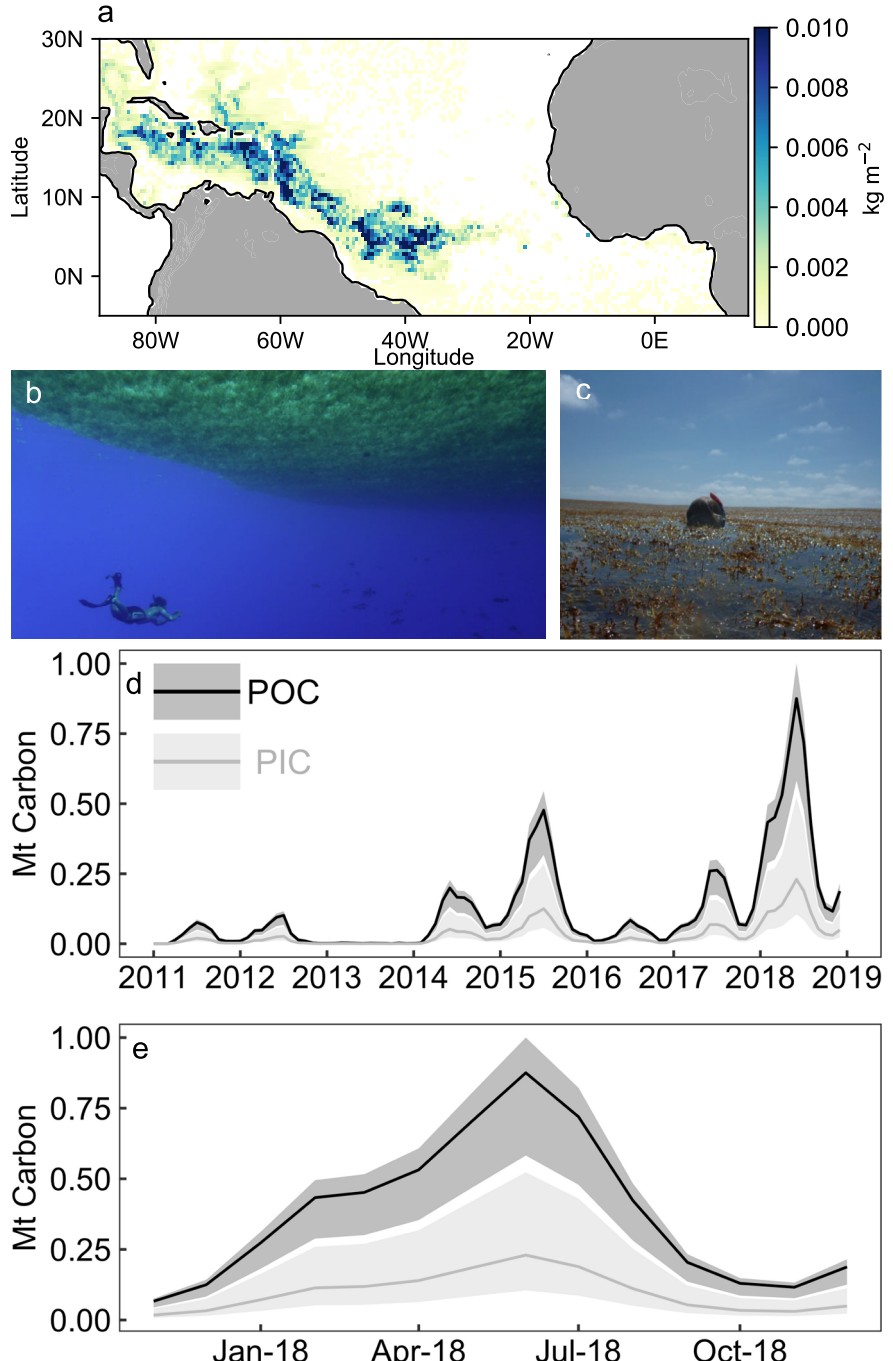

**Fig. 1 Floating *Sargassum* biomass during the 2018 Great Atlantic *Sargassum* Belt (GASB) event. a** A satellite image of *Sargassum* wet weight biomass in the North Atlantic during June 2018[13]. **b**, **c** Illustrative pictures of *Sargassum* rafts in the Sargasso Sea (2013) with a diver underneath or inside a raft for perspective. **d**, **e** Cumulative *Sargassum* particulate organic carbon (POC) and particulate inorganic carbon (PIC) within 5°S–25°N, 89°W–15°E from 2011 to 2018 or the 2018 growth cycle. The shaded areas indicate upper and lower bounds depending on *Sargassum* PIC:POC ratios.

epibiont calcification. The average C:N ratios (mol:mol, range in parentheses) of other brown, red, and green seaweeds are 19 (8–55), 19 (6–78), and 14 (7–24), respectively[28], indicating that *Sargassum* is within the higher C:N range of seaweeds and closer to the upper bound of CDR$_{theoretical}$ achievable with ocean afforestation (see also Supplementary discussion 1).

Another addition to CDR$_{theoretical}$ by *Sargassum* is the photosynthetic formation of dissolved organic carbon (DOC) and its subsequent release into seawater[29]. Indeed, *Sargassum* may have produced ~1 Mt DOC during the 2018 GASB which

exceeds the build-up of POC of 0.81 Mt C (Supplementary Table 1). However, seaweed-DOC mixes with the background DOC pool, is displaced by ocean currents, and generally only ~2% of DOC escapes remineralization for ≥20 years[30]. Therefore, accounting for CDR via DOC requires dedicated monitoring of vast ocean volumes with in situ sensors, that can differentiate DOC sources. So long as such monitoring is unfeasible, DOC may not be considered in CDR$_{theoretical}$ because accountability and independent verification of DOC storage are likely essential to finance ocean afforestation via carbon trading[31–34].

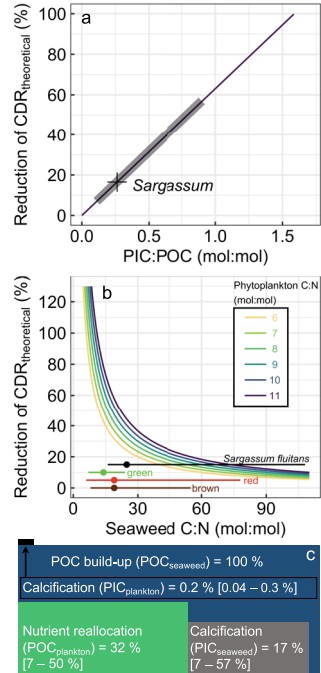

**Fig. 2 Reduction of the theoretical CO₂ removal potential (CDR_theoretical) through biogeochemical feedbacks.** a Reduction of CDR$_{theoretical}$ through particulate inorganic carbon (PIC) production by epibiont calcifiers associated with particulate organic carbon (POC) formed by *Sargassum*. The purple line shows the general relationship, the grey shaded part the range of *Sargassum* PIC:POC observations, and the cross the *Sargassum* mean. b Reduction of CDR$_{theoretical}$ due to nutrient reallocation, which becomes more pronounced the more the seaweed carbon-to-nitrogen (C:N) ratio approaches the phytoplankton C:N ratio. We used a range of phytoplankton C:N ratios[22] as indicated by the colour code. The horizontal lines display the range of C:N in *Sargassum*[72] and green/red/brown seaweeds, respectively[28] (the height of the lines on the y-axis has no meaning). The solid symbols on the horizontal lines are averages (*Sargassum* average = 24.8). c Summary of discounts and additions to CDR$_{theoretical}$ due to calcification and nutrient reallocation following Eq. 1 (upper and lower bounds in square brackets). Percent reductions due to POC formed by plankton (POC$_{plankton}$, blue); Percent reduction due to PIC associated with seaweeds (PIC$_{seaweed}$, grey); Percent increase due to PIC associated with plankton (PIC$_{plankton}$, black).

**Multiple challenges to verify CDR by ocean afforestation.** To this point, we have quantified how much CO₂ dissolved in seawater is fixed by *Sargassum* during the 2018 GASB (i.e., CDR$_{theoretical}$). Ultimately, however, we need to quantify the permanent influx of atmospheric CO₂ into the oceans after afforestation has generated a seawater CO₂ deficit. As a first step, this requires traceable air-sea CO₂ transfer following CO₂ fixation by seaweeds[8]. We calculated air-sea equilibration timescales ($\tau_{CO2}$) for the locations of pronounced *Sargassum* occurrence in the GASB region following Jones et al. (see ref. [35]) and found that these range between 2.5–15 months (mean = 5; Fig. 3). This is 2.5–18 (mean = 5.5) times longer than the modelled residence time of seawater in the surface mixed-layer ($\tau_{res}$) over much of the GASB region, which is between 0.3–1.5 months (mean = 0.9; Fig. 3b, c; see ref. [35]; but note that large datagaps occur in the Intra-Americas Sea). For the entire GASB region between 5°S–25°N and 89°W–15°E[13], $\tau_{CO2}$ is 2–46 (mean = 8) times longer than $\tau_{res}$, which is between 0.2–2 months (mean = 0.9; Fig. 3b, c; see ref. [35]). Hence, seawater containing a CO₂-deficit generated through afforestation has a high chance to lose contact with the atmosphere before the deficit is fully replenished with

atmospheric CO₂ (i.e., before equilibration is complete). Complete equilibration will likely occur some time in the future when seawater is eventually reexposed long enough to the surface but this may take >100 years[8]. The potential time-lag raises the question at what time-point can CDR$_{theoretical}$ be considered to be realized, because CO₂-fixation and seawater CO₂-absorption can be spatially and temporally uncoupled. In practice, CDR will likely only be rewarded in a carbon trading system when atmospheric CO₂ uptake is accountable and independently verifiable[31]. Hence, a key challenge for the implementation of ocean afforestation (and other marine CDR methods) on a carbon trading market is to quantify the CO₂ influx into the ocean, as this requires sophisticated measurements of seawater CO₂ uptake over large spatial and temporal scales.

After quantifying air-sea CO₂ transfer, seaweed carbon must be stored in a reservoir isolated from the atmosphere. The two storage methods widely anticipated are (i) biomass combustion followed by underground CO₂ storage (BECCS) or (ii) biomass deposition on the deep seafloor[1]. Underground storage in appropriate geological formations comes with additional offsets to CDR$_{theoretical}$, for example through shipping *Sargassum* biomass (0.0014–0.017% t wetweight$^{-1}$ km$^{-1}$) or through CO₂ separation after combustion (e.g., ~10–20% in the case of BECCS). Seafloor deposition may be associated with less process-related offsets to CDR$_{theoretical}$, when the deep sea is adjacent to the afforestation site. The value of geological and sea floor deposition depends upon a yet to be established policy framework that needs to clarify how sequestration permanence is factored into a carbon price, and how stringently CO₂ leakage back into the atmosphere must be monitored[36]. Geological sequestration in appropriate formations can be considered long-term with a 66–90% chance of less than 1% CO₂ leakage over 1000 years and possibly beyond[37]. Seafloor deposition will be less permanent than that. *Sargassum* sinks at 2500 m/d when its flotation bladders are removed[38], so that there is little time for degradation until reaching the seafloor (see also Supplementary discussion 2). However, on the deep seafloor and within its sediments, >90% of deposited carbon is typically remineralized and released back into the water column[25,29]. Thus, a significant fraction will eventually be transported back to the surface as respired CO₂ by ocean circulation and mixing. Data-constrained modelling suggests that the zonally-averaged mean time from deep seafloor remineralization to re-exposure to the atmosphere ranges between 700–900 years in the North Atlantic[39], where GASB biomass is deposited[38], and increases further along the global conveyor belt to >1400 years in the North Pacific[39]. These timescales are generally longer than those considered for some land-based CDR methods like terrestrial afforestation or soil carbon sequestration, which are on the order of decades to a century[36,40]. Nevertheless, the systematic differences reveal that some locations are better suited for seafloor deposition of seaweed biomass than others. The necessary monitoring of inevitable CO₂ leakage appears at least equally challenging as the monitoring of air-sea CO₂ uptake (see previous paragraph), because relatively small amounts of respired CO₂ must be traced in a vast volume and be distinguished from the background CO₂ pool. This monitoring challenge will likely influence the feasibility to implement seafloor seaweed deposition in carbon trading.

**Albedo climate feedbacks of ocean afforestation.** In addition to CO₂-related feedbacks on green-house radiative forcing, afforestation also influences radiative forcing by changing the Earth's albedo, the ratio between reflected and incident solar flux at the Earth surface. Afforestation in terrestrial environments reduces the albedo because forests are usually darker and reflect less

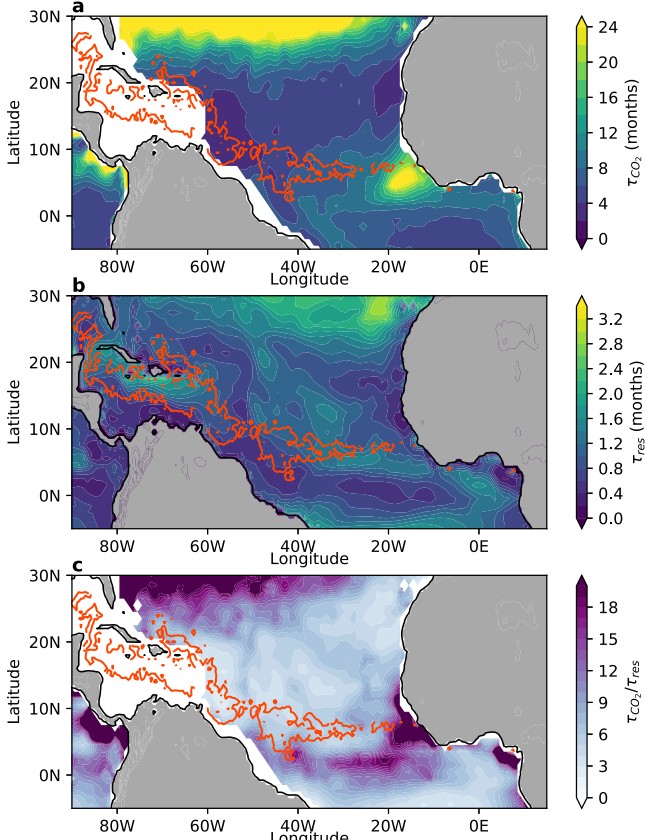

**Fig. 3 Timescales of CO₂ equilibration in the Great Atlantic *Sargassum* Belt.** The orange mask indicates the occurrence of *Sargassum* with >0.002 kg/m² during June 2018 as in Fig. 1a. **a** Annual mean timescales of $CO_2$ equilibration ($\tau_{CO_2}$) between the ocean and atmosphere. **b** Residence time of seawater in the surface mixed layer ($\tau_{res}$) before losing contact with the atmosphere. **c** Ratio of $\tau_{CO2}$ and $\tau_{res}$.

short-wave radiation than other landscapes[41]. This negative feedback can substantially reduce or even offset the cooling potential of terrestrial afforestation generated by CDR[41–43]. In the case of ocean afforestation, albedo increases because seaweeds reflect more short-wave radiation than seawater, especially when they occur near the sea surface[44]. We estimate that (sub)tropical ocean afforestation of the area of the 2018 GASB (~6100 km²) would reduce radiative forcing in the range of 180 (seaweed at ~5 m depth) to 1800 (seaweed floating at the surface) peta J/y. This exceeds an estimated reduction of radiative forcing of 42 peta J/y caused by 0.42 million tonnes of CDR_theoretical. Accordingly, the magnitude of these numbers suggest that ocean afforestation would also need to be assessed in the context of solar radiation management, i.e., the deliberate reflection of solar radiation into space. However, ocean afforestation is typically envisioned with benthic seaweeds cultured several meters below the surface to avoid storm damage[1,4]. Such culturing practices would alleviate the albedo effect[44]. Furthermore, our estimates do not consider indirect albedo feedbacks, such as the release of bioaerosols, which may counteract or enhance the albedo effect[45]. Indeed, identifying and accounting for all albedo feedback mechanisms induced by ocean afforestation will likely be as challenging as the comprehensive assessment of CDR (see Supplementary discussion 3).

**Synthesis and ramifications.** The GASB analysis sets realistic bounds on ocean afforestation CDR and provides the first

estimates of the relevance of the direct albedo effect. Two biogeochemical feedbacks, calcification and nutrient reallocation, reduce the theoretical CDR potential of ocean afforestation. Propagating both the upper and lower bounds for these feedbacks indicates that the CDR_theoretical of the 2018 GASB ranges from −0.03–0.8 Mt C. Accordingly, ocean afforestation at the scale of the GASB could constitute a net $CO_2$ source or, at best, contribute 0.0001–0.001% to the amount of annual CDR required in 2100 under a low emission scenario. The large range of estimates— from positive to negative—underlines that CDR with ocean afforestation is heavily dependent on feedbacks with the Earth System, which need to be further constrained. In addition, our analysis confirms that verification of both air-sea $CO_2$ flux and the permanence of carbon storage remain major challenges. Thus, significant physical and chemical oceanographic questions must be answered, along with establishing governance rules on verification and permanence to fully evaluate ocean afforestation. Moreover, the influence of ocean afforestation on ocean albedo could be more relevant than the climatic effect of CDR. However, albedo enhancements are currently not considered within carbon trading and thus would need additional legislation to be incentivised and rewarded[46]. Last, all the abovementioned feedbacks on the net climatic effect must also be evaluated for an environment that is constantly being modified due to ongoing climate change.

The assessment of terrestrial afforestation is further advanced than that of ocean afforestation, because ubiquitous natural analogues (forests) provide constraints on its efficacy. Thus, there is substantial knowledge of how Earth-system feedbacks modify the apparent CDR potential of terrestrial afforestation (see the wider discussion around ref. [47]). The GASB provides a new opportunity to constrain ocean afforestation, and our analysis points towards a similar conclusion as for terrestrial afforestation; i.e., climate intervention through ocean afforestation must consider complicated Earth-system feedbacks (Fig. 4), which could influence the sign and magnitude of its cumulative climatic effect.

Acceleration in CDR research and development is urgently needed as global negative emissions must be upscaled to gigatonnes within this decade (Supplementary Fig. 2). This requires rapid identification of tractable CDR solutions, which enable accurate and ongoing accounting of their overall climatic influence. The GASB analysis reveals that the net climatic impact of ocean afforestation is associated with major uncertainties, largely due to the inherent complexity of biological systems (Fig. 4). Other marine biological CDR approaches such as ocean iron fertilization exhibit similar complexity, and two decades of investigation has revealed comparable uncertainties about their net climatic impact. These ongoing unknowns raise the question of whether CDR using marine biota has the potential to be sufficiently well understood within this decade to prioritize development. Instead, the complexity associated with such CDR approaches may provide a compelling argument to focus on bottom-up engineered and better understood abiotic methods.

## Methods

**Build-up of seaweed biomass in the Great Atlantic *Sargassum* Belt (GASB).** Floating *Sargassum* biomass in the GASB has been quantified with satellites[15,48]. We downloaded monthly mean cumulative *Sargassum* wet weight data for the GASB region (Caribbean Sea and Central Atlantic Ocean) from ref. [16] under the following link (https://www.ncei.noaa.gov/access/metadata/landing-page/bin/iso?id=gov.noaa.nodc:0190272#). The cumulative wet weight biomass was converted to total particulate carbon (TPC) by multiplication with the wet weight to TPC conversion factor of 0.0543 (g:g) given in the ref. [49]. Supplementary Table 1 provides wet weights[16] and corresponding TPC values for the 2018 growth cycle, which commenced in November 2017 and constitutes our natural analogue case study for ocean afforestation.

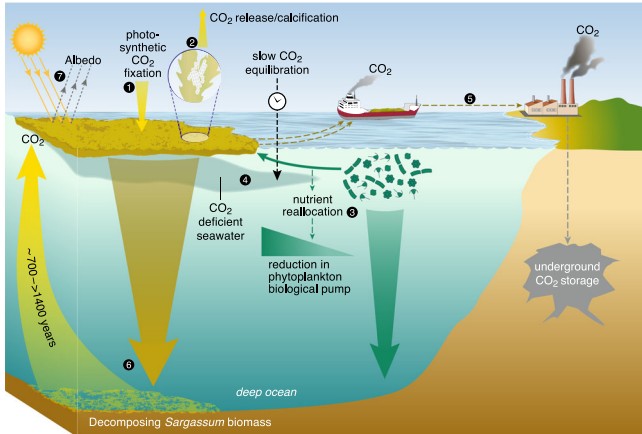

**Fig. 4 Summary of feedbacks discussed in this study.** The numbers in the black circles denote individual processes in the order they were discussed. 1. Photosynthetic carbon fixation by *Sargassum* consumes $CO_2$ dissolved in seawater. 2. *Sargassum* provides habitat for epibiontic encrusting organisms to calcify, thereby generating $CO_2$ through complicated feedbacks in the seawater carbonate system. 3. Nutrients taken up by *Sargassum* become unavailable for phytoplankton thereby reducing the natural carbon sequestration by phytoplankton. 4. $CO_2$ deficient seawater (driven by photosynthesis) can be subducted below the surface layer before it fully equilibrates with atmospheric $CO_2$. 5. *Sargassum* biomass can be harvested and transported to shore and used for Bioenergy with Carbon Capture and Storage (BECCS) causing $CO_2$ discounts. 6. Alternatively, *Sargassum* could be deposited on the seafloor but a large fraction of the respired *Sargassum* carbon would come back to surface on centennial-millenial timescales. 7. *Sargassum* increases albedo at the sea surface but could also induce complicated indirect albedo effects of unknown sign and magnitude.

**Calcification by epibionts living on *Sargassum*.** The precipitation of calcium carbonate (i.e., $Ca^{2+} + CO_3^{2-} \rightarrow CaCO_3$) reduces seawater alkalinity, thereby shifting the carbonate chemistry equilibrium and increasing the concentration of $CO_2$ (see refs. [18,50]). Thus, calcification by epibiont calcifiers counteracts photosynthetic $CO_2$ uptake by *Sargassum* and the desired CDR due to seaweed growth[19]. In the following, we go step-by-step through our calculation (and associated assumptions) to estimate the importance of the calcification-related offset by using the GASB as a natural analogue.

Wang et al. (see ref. [49]) have measured wet weight and total particulate carbon (TPC) content of *Sargassum* for their GASB biomass analysis (Supplementary Table 1). They have not separated TPC into fractions of particulate organic carbon (POC) and particulate inorganic carbon (PIC, i.e., carbonates). *Sargassum* does not form PIC itself but it provides substrate for calcifying epibionts (e.g., bryozoans or polychaete worms like *Spirorbis* spp.), which form firmly-attached calcareous structures on seaweed tissues[51,52]. $CaCO_3$ formed by these epibionts can constitute between 4.3–21.4% of the *Sargassum* wet weight biomass, with the average being 9.4% over the course of the year[20]. We used this average wet weight percentage to calculate the $CaCO_3$ mass of the GASB based on total wet weights as:

$$CaCO_3 \text{ mass} = \text{wet weight} * 0.094, \quad (2)$$

where the wet weight values were GASB July averages from 2011 to 2017 reported by Wang et al. (see ref. [49]). This range yields $CaCO_3$ masses from 0.0032 to 0.415 Mt $CaCO_3$, which were converted into mols by division with its molecular weight (i.e., 100 g/mol). (There is uncertainty associated with this as calcifying epibionts may also precipitate $MgCO_3$ with a lower molecular weight (i.e., 84 g/mol). We neglect this uncertainty because we have no information about the fraction of $MgCO_3$, and assuming all carbonates to be $CaCO_3$ sets a lower bound to the CDR offset through calcification. Assuming all carbonates were $MgCO_3$ would increase the offset further because the formation of 1 g of $MgCO_3$ reduces alkalinity slightly more than 1 g of $CaCO_3$. The calculation provided the mols of PIC associated with *Sargassum* which were subsequently subtracted from the mols of TPC reported by Wang et al. (see ref. [49]) to calculate POC. This enabled us to calculate a PIC:POC ratio (mol:mol) of 0.265 for *Sargassum* and the associated epibiont biomass under the assumption that 9.4% of the wet weight is $CaCO_3$ (see above). For the range of $CaCO_3$ wet weight percentages reported by Pestana (4.3–21.4%)[20], PIC:POC ranges from 0.11 to 0.9 (mol:mol).

The next step was to calculate how many moles $CO_2$ would be formed per mol PIC precipitated by the epibionts. This ratio, known as psi, was calculated following Frankignoulle et al.[18] using the psi-function of the carbonate chemistry calculation software seacarb[53] for R with recommended default settings for carbonate chemistry constants[53] (e.g., K1 and K2 from ref. [54]). For this calculation, we assumed subtropical conditions with a total alkalinity of 2350 µmol kg$^{-1}$, a dissolved inorganic carbon concentration of 2047.5 µmol kg$^{-1}$, a salinity of 35, and a temperature of 25 °C (the following code was used: psi < - psi(flag = 15, var1 = 2350e−6, var2 = 2047.5e−6, S = 35, T = 25, P = 0, Pt = 0, Sit = 0, pHscale = "T", kf = "pf", k1k2 = "l", ks = "d"). In this case psi is 0.63.

Finally, we multiplied psi with the mols of PIC precipitated by *Sargassum* epibionts and compared the amounts of generated $CO_2$ with the amounts of fixed $CO_2$ by photosynthetic POC production:

$$\text{Calcification offset} = \frac{\text{PIC*psi}}{\text{POC}}. \quad (3)$$

According to this, the calcification offset is 0.165 or 16.5% to the amount of POC fixed photosynthetically by *Sargassum*. The key assumption in this calculation is that 9.4% (ranging from 4.3 to 21.4%)[20] of *Sargassum* wet weight is $CaCO_3$, which is based on measurements with *Sargassum* collected in the Sargasso Sea[20]. These values provide a useful first estimate but they may differ for *Sargassum* occurring in the GASB since 2011, or for other species usef for ocean afforestation as pointed out in the discussion.

**Nutrient reallocation from resident phytoplankton to *Sargassum*.** Ocean afforestation requires macro-nutrients (Nitrogen (N) and Phosphorous (P)) and micro-nutrients such as iron (Fe), which are the primary limiting resources for primary production in the subtropical Atlantic[21]. Nutrients fixed by afforested seaweeds in offshore environments can no longer support carbon fixation by phytoplankton. Thus, nutrient reallocation from phytoplankton to seaweeds reduces phytoplankton carbon sequestration, a natural marine $CO_2$ sink. The phytoplankton carbon sequestration that would have been possible with the real-located nutrients must be subtracted from the CDR potential of ocean afforestation. In the following paragraphs, we go through the calculations and considerations made to assess the degree of reduction in seaweed CDR through nutrient reallocation.

The 2018 *Sargassum* bloom in the GASB commenced in November 2017 and peaked in June 2018 with a net biomass build-up of 18.8 Mt wet weight (Supplementary Table 1). Wet weight was converted into Mt N by multiplication with the *Sargassum* N:wet weight ratio (g:g) reported in Wang et al. (~0.002)[49]. Next, the mass of N was converted into gigamole (Gmol) by division with the molecular weight of N (i.e., 14.007 g/mol). The same calculation was done for P using the P:wet weight ratio of 0.0002 (g:g)[49] and the molecular P weight of 30.974 g/mol. Fe removal was calculated the same way (molecular weight of 55.845 g/mol), but we had to derive a Fe:wet weight ratio from other references. We used Fe: wet weight ratios of $1.009 \times 10^{-5}$ (g:g) determined for the benthic *Sargassum ilicifolium* from the Red Sea[55] (please note that Anton et al. (see ref. [55]) provide Fe: dry weight ratios but also give a wet weight:dry weight ratio of 8.18, which we use for the conversion). While this may not be fully representative for holopelagic *Sargassum* from the GASB, it may provide a useful number within a plausible range.

Based on these calculations, *Sargassum* growth during the 2018 GASB consumed 2.7, 0.12, and 0.003 Gmol N, P, and Fe, respectively. In the sub(tropical), Atlantic phytoplankton is mostly N-limited[21]. Assuming a typical organic matter C:N ratio of 8 in the (sub)tropical Atlantic[22], phytoplankton could fix 0.26 Mt C with the amount of N bound in *Sargassum*.

The C:N ratio for *Sargassum* as determined by Wang et al. (see ref. [49]) is 31.3 (mol:mol). However, this value includes PIC from epibionts which needs to be subtracted assuming a wet weight percentage of $CaCO_3$ of 9.4% (see previous section). With this correction the *Sargassum* organic C:N is 24.8 (i.e., (TPC-PIC)/N, where TPC and N is from Wang et al. (see ref. [49]) and PIC calculated as described in the previous section) and 28.2–16.5 for wet weight $CaCO_3$ percentages from 4.3 to 21.4% (see ref. [20]). Thus, phytoplankton are able to fix around one third (i.e., 8/24.8 × 100 = 32%) of the carbon fixed by *Sargassum* seaweeds (or 28–48% for the range of $CaCO_3$ wet weight percentages). The 32% must be subtracted from the CDR potential of afforestation, because phytoplankton carbon fixation with these nutrients would likely have occurred under the absence of the GASB. To test whether this percentage offset is sensitive to the assumption of N-limitation in the sub(tropical) North Atlantic[21], we repeated the calculation assuming P-limitation. The molar C:P ratio of *Sargassum* is 550 (i.e., (TPC-PIC)/P where TPC and P is from Wang et al. (see ref. [49]), and PIC calculated as described in the previous section) compared to 170 typically found in oligotrophic plankton communities[23]. Thus, as for N-limitation, phytoplankton are able to fix around one third (i.e., 170/550 × 100 = 31%) of the carbon fixed by *Sargassum* seaweeds when assuming P-limitation (Supplementary Fig. 1) (or 26–45% for the range of $CaCO_3$ wet weight percentages). As the differences between N-limitation or P-limitation are marginal, we focussed on C:N ratios in our study.

**Dissolved organic carbon (DOC) production.** Powers et al. (see ref. [56]) conducted incubation experiments with *Sargassum* collected in the Sargasso Sea under in situ

light and temperature conditions, and reported DOC release rates of $288 \pm 24$ µg C $d^{-1}$ $g_{wetweight}^{-1}$. Multiplying their daily rate with satellite derived biomasses yields a cumulative *Sargassum* DOC production for Nov. 2017—Dec. 2018 of 1.07 Mt C (Supplementary Table 1). Please note that this estimate is associated with several uncertainties that are difficult to quantify[56]. While the uncertainty of the reported rate is relatively minor (SD ± 24 µg C $d^{-1}$ $g_{wetweight}^{-1}$), it has been determined only during a certain time of the year (September) when *Sargassum* was in a distinct growth stage. As mentioned by Powers et al., (see ref. [56]) DOC production rates may be different for early or late season *Sargassum*, and also change with the environmental conditions such as light and temperature.

**Air-sea $CO_2$ exchange and equilibration timescales**. To generate atmospheric $CO_2$ removal, the $CO_2$ deficit generated through afforestation needs to be balanced by atmospheric $CO_2$ influx into the ocean. Air-sea $CO_2$ fluxes in the open ocean primarily depend on wind speeds (U) and the air-sea $pCO_2$ difference. Temperature (T) and salinity (S) also play a role as they influence the Schmidt number (Sc) and the solubility (K) of $CO_2$ in seawater[57]. We followed the approach by Jones et al. (see ref. [35]) to calculate the timescale of air-sea equilibration ($\tau_{CO_2}$) as:

$$\tau_{CO_2} = \frac{h \times R}{G \times B}, \tag{4}$$

where $h$ is the mixed layer depth (in $m$), $R$ is the ratio of dissolved $CO_2 + H_2CO_3$ to total dissolved inorganic carbon (DIC), $G$ is the gas transfer velocity (in m/s), and $B$ is the Revelle Factor. In a first step, we calculated $\tau_{CO_2}$ for a range of plausible scenarios for (sub)tropical environments, where ocean afforestation could be applied. In these scenarios, we assumed typical mixed layer depths ranging from 10 to 110 m[58]. $R$ and $B$ were calculated with the seacarb package using the carb and buffergen functions with recommended default settings for equilibrium constants[53] (e.g., K1 and K2 from ref. [54]). As input variables we used a total alkalinity (TA) of 2350 µmol/kg[59] and assumed equilibrated $pCO_2$ of 410 µatm. Nutrients (P and Si) were set to zero and salinity to 35. Temperature varied between 20–30 °C depending on the scenario (see Supplementary Fig. 3). $G$ was calculated following the empirical equation by Wanninkhof[60] as:

$$G = 0.251 \times U_{10}^2 \times \left(\frac{Sc}{660}\right)^{-0.5}, \tag{5}$$

where $U_{10}$ is the wind speed 10 m above ground and Sc is the Schmidt Number. By convention, $G$ (typically abbreviated as $k$) is reported in cm/h and $U_{10}$ in m/s. Thus, the coefficient 0.251 has the unit $(cm\ h^{-1})\ (m/s)^{-2}$. Sc is temperature-dependent and was calculated as in ref. [60]. $G$ was converted from cm/h to m/s by dividing it with 360,000.

Supplementary Fig. 3 shows $\tau_{CO_2}$ for a range of scenarios under (sub)tropical conditions. $h$ and $G$ have the largest influence on $\tau_{CO_2}$ as has been pointed out previously[35]. The driving force behind $G$ is the wind speed while increasing $h$ increases $\tau_{CO_2}$, because a larger volume of water needs to be equilibrated (Supplementary Fig. 3A). Temperature influences $\tau_{CO_2}$ through its influence on $R$, $B$, and Sc but its effect is small relative to h and wind speed (Supplementary Fig. 3B).

To further explore the constraints of air-sea $CO_2$ exchange on CDR, we mapped $\tau_{CO_2}$ for the GASB region using observational datasets and ERA5 wind data products. $R$ and $B$ were calculated with the Python script PyCO2SYS version 1.0.1 by Matthew Humphreys et al. (see ref. [61]) using the Takahashi et al. monthly climatology[59] for TA and $pCO_2$. Input values for phosphate, silicate, temperature, and salinity were also taken from the Takahashi et al. climatology[59], with all variables on a 4° latitude by 5° longitude grid. We used the default constants recommended for seacarb[53] (see above) also for the calculations with PyCO2SYS (e.g., K1 and K2 from ref. [54]).

Mixed layer depth ($h$) is calculated from a 1° latitude by 1° longitude gridded Argo climatology[62] of temperature and salinity provided as monthly means, averaged for the time period 2004 through 2018. Mixed layer depth is defined using a density threshold of 0.03 kg m$^{-3}$ following de Boyer Montégut et al. (see ref. [63]). Finally, $G$ is calculated as in Eq. 5, using gridded Argo temperature to determine Sc following Wanninkhof (see ref. [60]), and ERA5 monthly mean wind speeds for the year 2018 for $U_{10}$. ERA5 winds, $R$, and $G$, are linearly interpolated onto the Argo 1° latitude by 1° longitude grid to compute $G$ and $\tau_{CO_2}$. We produce an annual mean estimate of $h$, $R$, $B$, $G$, and $\tau_{CO_2}$ by averaging the monthly mean Argo and ERA5 data, noting that the time periods are different for these data and the Takahashi climatology[59], with the interpolated annual mean components shown in Supplementary Fig. 4. Additionally, we make seasonal estimates of $\tau_{CO_2}$ by averaging the monthly data for all variables to 3-month means (Supplementary Fig. 5).

The $\tau_{CO_2}$ timescales calculated for the GASB region (Fig. 3 main text; Supplementary Fig. 5) are generally in good agreement with those calculated by Jones et al. and both analyses show the same regional features (compare Fig. 3 main text with Fig. 1a in the ref. [35]). Nevertheless, our $\tau_{CO_2}$ timescales tend to be slightly longer, especially north of 25°N. We used the same underlying datasets for $h$, $R$, and $B$ so these factors cannot explain the differences. However, we used a more recent parameterization of air-sea gas exchange, which increases $G$ by about 20% relative to the one used by Jones et al. Furthermore, Jones et al. used a wind speed climatology based on a monthly mean QuikSCAT and National Centers of

Environmental Prediction (NCEP) reanalysis averaged from January 2000 to July 2007 whereas we used ERA5 data for 2018. Sensitivity calculations show that $\tau_{CO_2}$ timescales are highly dependent on wind speed, especially in the critical range below 10 m/s (Supplementary Fig. 3A).

The residence time of seawater in the surface mixed layer was provided by Jones et al. (see ref. [35]) from their simulation of an idealized surface age tracer using circulation fields from the Estimating the Circulation and Climate of the Ocean-Global Ocean Data Assimilation Experiment product, version 3 iteration 73, a 1° × 1° global data assimilating product based on the Massachusetts Institute of Technology General Circulation Model[35]. All relevant details on their calculations and assumptions can be found in their paper (see Section 2.2 in ref. [35]).

**$CO_2$ offset through biomass transportation and processing associated with geological carbon storage**. Seaweed carbon storage in geological reservoirs requires energy for transportation from offshore seaweed farms to the processing and storage sites. Biomass could be transported as wet weight, dry weight, or as some further refined product[64]. Storage could be in various depots and in various physical states such as biochar or liquid $CO_2$ (see refs. [65,66]). It is beyond the scope of this study to go through all possible scenarios and estimate the associated $CO_2$ discounts for each of them in a full lifecycle analysis. We therefore consider possible $CO_2$ discounts with two central steps (shipping, $CO_2$ separation from flue gas stream) within one plausible scenario, where harvested seaweed wet weight biomass is transported from the oceans to the coast and combusted for bioenergy with carbon capture and storage (BECCS). Please note that we neglect possible $CO_2$ emissions during harvesting, drying, (un)loading as well as other $CO_2$ offsets associated with BECCS such as liquified $CO_2$ transports.

$CO_2$ emissions by bulk carriers (ships transporting unpackaged goods such as coal) depend on the vessel size but range from 2.7 to 33.9 g $CO_2$ t cargo$^{-1}$ km$^{-1}$ with the higher value for smallest vessels[67]. One tonne of *Sargassum* wet weight contains 54,300 g of carbon[49], equivalent to 200,000 g of $CO_2$ fixed as carbon in biomass. Thus, the $CO_2$ discount through shipping is 0.0014–0.017 % t wetweight $^{-1}$ km$^{-1}$, depending on the vessel size. Transport from an offshore macroalgae farm 100 km away from the processing facility would therefore reduce $CDR_{theoretical}$ by 0.14–1.7% due to combustion of engine fuels. This sets a lower bound since energy consumption during (un)loading as well as fuel production and ship construction are not considered.

For BECCS, $CO_2$ generated during macroalgae combustion needs to be separated from the flue gas, which has an efficiency of 80–90% depending on the applied method[37,66]. Accordingly, losses of $CO_2$ during the separation process would reduce $CDR_{theoretical}$ by 10–20%. The energy for processing the $CO_2$ for storage could theoretically be derived from combusting the algae, but there are still other potential $CO_2$ offsets along the process chain (see above and ref. [66]). Thus, the 10–20% discount considered for BECCS is likely another lower bound of the potential offsets to $CDR_{theoretical}$.

**Albedo modification by *Sargassum***. Albedo is defined as the ratio between reflected and incoming solar flux at the Earth surface[68]. It is given as a dimensionless number between 0 (absorption of all incident solar flux) and 1 (reflection of all solar flux). Seawater has a relatively low albedo compared to other surfaces[69]. Therefore, marine vegetation on top or slightly below the sea surface increases the albedo[44]. In the following, we estimate how much the albedo enhancement caused by afforestation could reduce radiative forcing. Afterwards, we estimate how much radiative forcing is reduced by CDR through afforestation. Last, we compare the reductions of radiative forcing. The calculations described below are based upon equations by Betts[41] and Kirschbaum et al.[43], who did a similar assessment in the context of terrestrial afforestation.

The 2018 GASB had initially a surface *Sargassum* coverage of ~870 km$^2$ in December 2017 (Supplementary Table 2). It then extended to a maximum of 6093 km$^2$ in June 2018 after which coverage declined to ~807 km$^2$ (Supplementary Table 2; refs. [13,16]). The seasonality in coverage as shown in Supplementary Table 2 prescribes the growth cycle for our ocean afforestation scenario, where we assume that (i) seaweed farms of 6093 km$^2$ are maintained north of South America within the GASB region (lat = 10°N; 60°W), (ii) growth occurs during approximately the first half of the year, and (iii) harvest/processing during the second half matching biomass build-ups and declines as in Supplementary Table 2. The (sub)surface growth of marine vegetation increases the albedo of seawater[44], thereby decreasing the daily average radiative forcing ($\Delta RF_{daily}$) calculated as:

$$RF_{daily} = Q_s \downarrow *\Delta a*(1 - \alpha_{atm}) \tag{6}$$

Here, $Q_s\downarrow$ is the total daily downward solar flux (in J m$^{-2}$ d$^{-1}$), $\Delta a$ the change in albedo over the shortwave spectrum (i.e., ~0.28–2500 nm) due to seaweed coverage, and $\alpha_{atm}$ is the proportion of shortwave-radiation absorbed by the atmosphere (~20%)[43,70]. $Q_s\downarrow$ data was obtained from the Giovanni online data system, developed and maintained by the NASA GES DISC. More specifically, we downloaded maps of monthly averages (December 2017–November 2018) of Surface incoming shortwave flux (0.5 × 0.625°) from the MERRA-2 Model M2TMNXRAD v5.12.4. (This data is provided in W m$^{-2}$ and was multiplied with 86,400 to convert to J m$^{-2}$ d$^{-1}$ needed for $Q_s\downarrow$ in Eq. 6.) Afterwards we extracted representative monthly incoming solar fluxes for the anticipated study region (lat = 10°N; 60°W). $Q_s\downarrow$ changes over the course of a seasonal cycle but is relatively

stable in the low latitude considered in our scenario, ranging from 180 to 280 W m$^{-2}$ (Supplementary Table 2; equivalent to 15.5–24 MJ m$^{-2}$ d$^{-1}$). The increase of albedo, $\Delta a$, due to *Sargassum* blooms has, to the best of our knowledge, not been determined so far. However, there are assessments of $\Delta a$ due to seagrass in shallow water published by Fogarty et al. (see ref. [44]). These authors found that albedo of seagrass meadows is by about 0.01–0.07 higher than open water albedo with the highest increase observed, when the coverage of the canopy was dense and close to the surface (the closest they considered was 0.25 m below surface)[44]. While not fully representative for our *Sargassum* analogue, Fogarty et al.'s data (see ref. [44]) provides a useful first indication how much seaweeds could increase albedo relative to open ocean seawater. Arguably, *Sargassum* rafts would be on the upper end or above of the 0.01–0.07 range, because they float right at the surface and have a very dense coverage[49].

Applying Eq. (6), we first calculate cumulative monthly $\Delta RF$ (i.e., $\Delta RF_{daily}$ * days of the month) and then cumulative yearly $\Delta RF$ (i.e., sum of monthly $\Delta RF$). We do this calculation for a range of plausible $\Delta a$ (0.01–0.1)[44], to account for different types of ocean afforestation where seaweeds could grow at different depths in the water column or even at the surface like *Sargassum* (Supplementary Tables 2 and 3). In all scenarios, $\Delta RF_{yearly}$ decreases ranging from 181—1811 PJ y$^{-1}$.

The reduction in radiative forcing per year$^{-1}$ due to the removal of 1 tonne of carbon from the atmosphere ($\Delta RF_C$ in J y$^{-1}$) can be calculated as:

$$\Delta RF_C = 86{,}400 \times 365 \times 5.35 \times \ln\left[\left(C_{atm} + \frac{1}{2.124 \times 10^9}\right)/C_{atm}\right] \times 510 \times 10^{12}, \quad (7)$$

where $C_{atm}$ is the $CO_2$ concentration in the atmosphere (we use 410 p.p.m.v. for our calculations), 86,400 is the number of seconds per day, 365 the number of days per year, $510 \times 10^{12}$ the surface area of the Earth, and $2.124 \times 10^9$ is the tonnes of carbon that lead to a 1 p.p.m.v. increase of atmospheric $CO_2$ (see ref. [43]). The factor 5.35 and the natural logarithm account for the increase of radiative forcing with increasing atmospheric $CO_2$ (see ref. [71]). According to Eq. 7, $\Delta RF_C$ equals −99 GJ tC$^{-1}$ y$^{-1}$. This number is slightly lower than −104 GJ tC$^{-1}$ y$^{-1}$ calculated by Kirschbaum et al.[43] because they used a $C_{atm}$ of 390 p.p.m.v. and the climate sensitivity is a logarithmic function of $C_{atm}$ (see ref. [71]).

The assessment shows that ocean afforestation increases albedo thereby constituting a positive feedback in addition to the reduction of radiative forcing caused by CDR. The CDR effect on radiative forcing, i.e., −99 GJ tC$^{-1}$ y$^{-1}$ × 420,000 t C = −41.6 PJ y$^{-1}$, is considerably smaller than the albedo effect during the first years (420,000 t C = CDR$_{theoretical}$ of the 2018 GASB; see main text). However, the reduction of radiative forcing through CDR is cumulative from season to season, so that it eventually becomes more important than the albedo effect. In our scenario seaweed farms totalling 6093 km$^2$ in the tropics, CDR would outweigh the albedo effect after −181/−41.6 to −1811/−41.6 (i.e., 4.4–44) annual seaweed growth cycles assuming instantaneous (and therefore unrealistic) atmospheric $CO_2$ invasion into seawater. Please note that this calculation of timescales neglects the change in $C_{atm}$ due to ongoing emissions and changes in atmospheric $CO_2$ as this has only a small influence on the relatively short timescales considered here.

**Propagating the upper and lower bounds for the reduction of CDR$_{theoretical}$ through calcification and nutrient reallocation.** In this section we propagate the wide range of assumptions for the calcification and nutrient reallocation offsets to estimate the upper and lower bounds of CDR$_{theoretical}$ of afforestation with *Sargassum*. The range of assumptions are constrained by published data as described in methods on calcification and nutrient reallocation. The goal of such propagations is to ensure that the best-case and worst-case scenarios for ocean afforestation are represented to provide a balanced viewpoint. Furthermore, we aim to illustrate the degree of uncertainty of the theoretical CDR potential of ocean afforestation due to these biogeochemical feedbacks. Our estimation is based on Eq. 1.

To estimate the lower bound of CDR$_{theoretical}$, we first assume an arbitrary amount of 100 mol TPC associated with *Sargassum*. Assuming the maximum of $CaCO_3$ contribution *Sargassum* wet weight by epibiontic calcifiers of 21.4%[20] yields an upper value PIC:POC ratio of ~0.9 (mol:mol). Thus, 100 mol TPC would be split in ~47 mol PIC and ~53 mol POC. The 53 mols of POC would bind the same amount of $CO_2$ whereas the formation of 47 mols PIC would release ~29.6 mols of $CO_2$ (i.e., PIC$_{seaweed}$ calculated as 47*psi). Accordingly, CDR$_{theoretical}$ would decrease from 53 to 23.4 mols $CO_2$ in this high calcification scenario. The 53 mols of POC would require ~3.3 mols of nitrogen assuming a lower bound C:N ratio of 16 (mol:mol) reported for holopelagic *Sargassum fluitans*[72]. These amounts of $N$ would support 26.5 mols of POC fixation by phytoplankton assuming a phytoplankton C:N ratio of 8 (see ref. [22]). Thus, CDR$_{theoretical}$ would decrease by another 26.5 mols (POC$_{plankton}$) and become slightly negative (i.e., 53 − 29.6 − 26.5 = −3.1). However, nutrient reallocation from phytoplankton to *Sargassum* would also reduce calcification by phytoplankton. The planktonic PIC:POC ratio in the (sub)tropical Atlantic is ~0.01 (see refs. [26,27]), so that 26.5 mols of POC would be associated with 0.265 mols of PIC. This phytoplankton PIC is not formed due to the nutrient reallocation to *Sargassum* which saves ~0.17 mols of $CO_2$ (PIC$_{plankton}$) from being released (0.265*psi). Overall, the CDR$_{theoretical}$ would be close to zero under

the abovementioned lower bound assumptions (i.e., $53 - 29.6 - 26.5 + 0.17 \approx -3$ mols for 100 mol TPC initially fixed by the *Sargassum* community). For the 2018 GASB with a production of ~1.02 Mt TPC (Supplementary Table 1), this would be −0.03 Mt C (formation of ~0.1 Mt $CO_2$).

To estimate the upper bound of CDR$_{theoretical}$ we do exactly the same calculations but with upper bound assumptions: $CaCO_3$ contribution *Sargassum* wet weight by epibiontic calcifiers of 4.3%[20] leading to a *Sargassum* PIC:POC of ~0.11 (mol:mol); C:N *Sargassum* = 108 mol:mol[72]; C:N phytoplankton = 8 (mol:mol, same as above); planktonic PIC:POC = 0.01 (mol:mol, same as above). Under these conditions, 10 of the initially 100 mol TPC formed in *Sargassum* habitats would be present as PIC and release 6.3 mol $CO_2$ (i.e., PIC$_{seaweed}$). The 90 mols of POC would require 0.83 mols of $N$. This would support 6.6 mol of phytoplankton POC (i.e., POC$_{plankton}$), which would form 0.066 mols of PIC thereby releasing 0.04 mols $CO_2$. Thus, the upper bound CDR$_{theoretical}$ is $90 - 6.3 - 6.6 + 0.04 \approx 77$ mols for 100 mol TPC initially fixed. For the 2018 GASB with a production of ~1.02 Mt TPC (Supplementary Table 1), this would be 0.79 Mt C (consumption of ~2.9 Mt $CO_2$).

**The amount of $CO_2$ removal to limit global warming below 2 °C, and the hypothetical contribution of the GASB to these $CO_2$ removal targets.** Results from simulations of shared socioeconomic pathways (SSPs) 1–5 with Integrated Assessment Models (IAMs) were downloaded in November 2018 from the database of the International Institute of Applied Systems Analysis (IIASA) under the link https://secure.iiasa.ac.at/web-apps/ene/SspDb/dsd?Action=htmlpage&page=about. The database was filtered for SSP scenarios with an additional temperature forcing of 2.6 W/m$^2$ (SSP-2.6) where global warming at the end of the simulation (i.e., 2100) was below 2 °C (transient temperature overshoots above 2 °C were not considered as an exclusion factor). This yielded 32 SSP-2.6 simulations. Afterwards, we examined each of the 32 simulations for the variable: Emissions | $CO_2$|Carbon Capture and Storage | Biomass (please note that IAMs currently realize $CO_2$ removal mostly with Bio Energy with Carbon Capture and Storage (BECCS) and terrestrial afforestation[73]). The time steps of IAMs shown here is 10 years (5 years from 2005 to 2010) so that the annual $CO_2$ removal was calculated by linear interpolation between two datapoints.

All IAM runs simulating SSPs included $CO_2$ removal, which generally commenced in the 2020's, although earlier in a few simulations (Supplementary Fig. 2). The integrated amounts of $CO_2$ removal from 2005 until 2100 were between 112–931 Gt $CO_2$ (minimum–maximum) with a median of 423 Gt $CO_2$ and a mean of 513 Gt $CO_2$. In general, $CO_2$ removal was lower in those scenarios that had a more ambitious emission reduction (e.g., SSP1; Supplementary Fig. 2).

Ocean afforestation at the scale of *Sargassum* growth in the GASB during 2018 could contribute −0.0001–0.0029 Gt $CO_2$ of $CO_2$ removal, if all of the seawater $CO_2$ consumed through biomass formation is balanced by permanent influx of atmospheric $CO_2$. The large range is due to propagating the upper and lower bounds of assumptions (see previous section) with respect to magnitude of calcification and nutrient reallocation.

## Data availability
Data used for calculations is provided in the supplementary material. In case calculations were based upon datasets from repositories, we provide the sources and/or necessary links in the methods or respective supplementary material.

## Code availability
All calculations, equations, and applied software are detailed and referenced in the methods and supplementary material.

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

## Acknowledgements

We thank Chuanmin Hu and Mengqiu Wang for providing their monthly composites of gridded *Sargassum* biomass data; Daniel Jones, Takamitsu Ito, Yohei Takano, and Wei-Ching Hsu for providing model output data on surface water residence time; Martin Jung and Jens Daniel Müller for their photographs of *Sargassum* rafts; Andrew Lenton for proof-reading the manuscript. This study was funded by the Australian Research Council by a Laureate awarded to P.W.B. (FL160100131). V.T. acknowledges support from CSHOR, a joint research Centre for Southern Hemisphere Ocean Research between QNLM and CSIRO. Argo data were collected and made freely available by the International Argo Program and the national programs that contribute to it (http://www.argo.ucsd.edu; http://argo.jcommops.org; https://doi.org/10.17882/42182). The Argo Program is part of the Global Ocean Observing System. The ERA5 dataset used for biomass and equilibration timescale estimations was developed by the European Centre for Medium-Range Weather Forecasts and was obtained from the Research Data Archive at the National Center for Atmospheric Research, Computational and Information Systems Laboratory. Analyses and visualizations used for albedo calculations were partially produced with the Giovanni online data system, developed and maintained by the NASA GES DISC. We acknowledge the mission scientists and Principal Investigators who provided the data used in the albedo calcuations.

## Author contributions

L.T.B., V.T., and P.W.B. designed the study. L.T.B. and V.T. analysed the data. L.T.B. drafted and revised the manuscript. P.W.B. and V.T. worked on the different versions of the manuscript. J.G., C.L.H., and J.A.R. commented on and contributed to the improvement of the manuscript.

## Competing interests

The authors declare no competing interests

## Additional information

**Peer review information** *Nature Communications* thanks Andreas Oschlies and other, anonymous, reviewers for their contributions ot the peer review of this work. Peer review reports are available.

