## [Peer Review File · Nature Communications]

REVIEWER COMMENTS

Reviewer #1 (Remarks to the Author):

I reviewed a version of this manuscript submitted to a different Nature family journal. The authors have done a satisfactory job of answering my review comments. I have nothing further and recommend publication.

Reviewer #2 (Remarks to the Author):

general comment:

The manuscript investigates the theoretical CDR potential of marine afforestation using the natural analogue of recently observed Sargassum blooms in the tropical Atlantic. The idea and concept of the manuscript are interesting and can provide much needed insights into accounting aspects and CDR potential of marine afforestation. Unfortunately, the manuscript gives the impression of trying to make the efficacy of marine afforestation appear as small as possible. e.g.

- (i) Sargassum might contain more CaCO₃ than other macroalgae,
- (ii) the nutrient robbing effect is computed based on N rather than P, the latter of which would arrive at a lower number,
- (iii) surface pCO₂ equilibration times are averaged over a too large area that includes overly high values. Averaging over the area of observed blooms (Fig.1) would yield lower equilibration times that are of similar magnitude as residence times (Fig.3c)
- (iii) changes in surface albedo are assumed to translate to changes in planetary albedo, which would be correct only on a cloud-free planet.

The comments regarding monitoring and accounting are interesting and useful, but offer a too pessimistic view on the possible economic incentives to deploy marine afforestation as CDR measure. Terrestrial afforestation has economic value and is considered in international carbon trading even though it would not pass the criteria on permanence and accountability rightly viewed problematic by the authors for marine afforestation.

I think a more balanced discussion is required to make this a useful scientific contribution to the debate.

individual points:

I.21-23 not correct: ocean afforestation is not generally seen as key component of the marine portfolio.

I.32 'alter...when upscaled' is not correct. First, the effects do not depend on scale, second, they don't alter the efficacy, but they determine the efficacy

I.90-92. the N:P ratio of consumed nutrients (22.5) is higher than that of ordinary phytoplankton (16). The draw-down of fixed N might thus be topped by nitrogen fixation, leading to a smaller effect of nutrient reallocation.

I.126-129. Why is permanent DOC storage essential to finance ocean afforestation? There is also no permanent C storage in afforestation on land, but still does not seem to affect funding or even consideration by UNFCCC.

I.135/136. The equilibration times shown in Fig.3 are high mostly north of 30N, but in the area of the

GASB marked in Fig.1 are generally smaller than 6 (color scale is not fully appropriate to resolve the western tropical Atlantic and Caribbean).

I.163 A storage time of 700-900 years would be much longer than anything that can be achieved by terrestrial afforestation, which is considered in current UNFCCC carbon accounting.

I.215 Ref.40 is an extremely poor example of knowledge on the CDR potential of terrestrial afforestation. It should not be cited in this positive tone. The related comments have shown that ref.40 was poorly researched and did not present a fair understanding of the carbon cycle.

I.281 Chung reference is incomplete
suppl. line 20 units of conversion factor missing.

S3 I.94ff impact of N₂ fixation is neglected that could offset much/all? of the nitrogen reallocation.

S8. Any change in sea surface albedo becomes effective for the planetary energy budget only in regions not covered by clouds. Changes in planetary albedo will in general be smaller than changes in surface albedo.

Dear Editor,

We thank the Reviewers for their thoughtful comments, which improved the manuscript. Please find our point-by-point responses below. Please note that line numbers refer to the revised version of the manuscript and supplementary materials.

Kind regards,
Lennart Bach

REVIEWER COMMENTS

Reviewer #1 (Remarks to the Author):

Comment 1 I reviewed a version of this manuscript submitted to a different Nature family journal. The authors have done a satisfactory job of answering my review comments. I have nothing further and recommend publication.

REPLY: We thank the Reviewer for the kind words and the constructive comments during the first iteration.

Reviewer #2 (Remarks to the Author):

general comment:

Comment 2 The manuscript investigates the theoretical CDR potential of marine afforestation using the natural analogue of recently observed Sargassum blooms in the tropical Atlantic. The idea and concept of the manuscript are interesting and can provide much needed insights into accounting aspects and CDR potential of marine afforestation.

REPLY: We thank the Reviewer for the kind words.

Comment 3 Unfortunately, the manuscript gives the impression of trying to make the efficacy of marine afforestation appear as small as possible. e.g.

(i) Sargassum might contain more CaCO₃ than other macroalgae,

(ii) the nutrient robbing effect is computed based on N rather than P, the latter of which would arrive at a lower number,

(iii) surface pCO₂ equilibration times are averaged over a too large area that includes overly high values. Averaging over the area of observed blooms (Fig.1) would yield lower equilibration times that are of similar magnitude as residence times (Fig.3c)

(iv) changes in surface albedo are assumed to translate to changes in planetary albedo, which would be correct only on a cloud-free planet.

REPLY: The goal of our analysis was to provide upper and lower bounds for the efficacy of ocean afforestation. These bounds are informed by data from the GASB. We have strived to be as neutral as possible. The impression by the Reviewer that we make it “appear as small as possible” may perhaps originate from relatively large offsets we estimated (20-100%). We

reviewed the text to eliminate any tone that could potentially be interpreted as biased as detailed in the following.

Re (i). It is correct that other algae may be associated with less (or more) CaCO_3 than the *Sargassum* literature values we used in our analysis (the literature on this topic is unfortunately very poor). We have already discussed this issue in the supplement but we agree that this is important and deserves to be addressed in the main text. We moved the following text from supplement 2 to the calcification paragraph in the main text: “It is currently unclear if 9.4% is applicable for the new *Sargassum* blooms occurring since 2011 in the GASB, or for other seaweeds potentially used for ocean afforestation. Slower/faster growing seaweeds may provide more/less time for epibiontic calcifiers to settle, which would affect the PIC:POC ratio accordingly. Over the range of wet weight CaCO_3 percentages reported for individual *Sargassum* samples from the Sargasso Sea (i.e. 4.3–21.4%)²⁰, the PIC:POC is 0.11-0.9 and the offset to CO_2 removal 7–57% (Fig. 2c; Supplement 2). This indicates that the calcification offset could range from being negligible to being a major factor reducing the CDR efficiency of ocean afforestation.” (see lines 90-97). These sentences provide a balanced view on the topic and hopefully reassure the Reviewer of our neutrality.

Re (ii): The choice to use N-limitation was based on findings that our study region ((sub)tropical North Atlantic) is primarily N-limited (Moore et al., 2013). To test whether the CDR discount is sensitive to the assumption of N-limitation in the sub(tropical) North Atlantic (Moore et al., 2013), we repeated the calculation assuming P-limitation (as suggested by the Reviewer). The molar C:P ratio of *Sargassum* is 550 (i.e. (TPC-PIC)/P where TPC and P is from Wang et al. (2018) and PIC calculated as in Supplement 2) compared to about 110-250 (with most observations around 170) typically found in oligotrophic plankton communities (Martiny et al., 2013). Thus, as for N-limitation, phytoplankton are able to fix around one third (i.e. $170/550 * 100 = 31\%$) of the carbon fixed by *Sargassum* seaweeds when assuming P-limitation. We added this estimation to the revised version of the supplementary material (lines 158-171), add a new figure which illustrates the P-based calculation (Fig. S1, see below), and point towards the outcome of the P-based estimate in the revised manuscript (line 111). The reason why the Reviewer assumes a lower CDR discount under P-limitation is likely due to the assumption of a Redfield phytoplankton C:P ratio of 106. However, Redfield only applies for rather nutrient rich (higher latitude) regions (from where it was originally derived), and not for the (sub)tropical Atlantic where C:P is substantially higher (Martiny et al., 2013). Due to the similar outcome for N- and P-limitation, we decided to stick to N-limitation, which is well justified based on findings by (Moore et al., 2013).

Figure S1. This figure has been added to the supplementary material to show that there is no substantial influence on the outcome of our study when assuming P instead of N-limitation. The figure has the same structure as Fig. 2 in the main text.

Re (iii): We agree with the Reviewer that this section needs a closer reference to the regions where *Sargassum* primarily occurs. Therefore, we added a mask to figure 3 that shows the detection of *Sargassum* ($> 0.02 \text{ kg/m}^2$) during July 2018 as in Fig. 1a. We changed the range of equilibration and residence timescales to the min-max range found within this mask. As pointed out by the Reviewer, equilibration times become lower (both the absolute value and relative to surface residence times of seawater). We changed the text to these lower numbers in the abstract (line 33) and in the main text (lines 153-158). However, we think it is also valuable to at least mention the larger range of values calculated for the entire map ($5^\circ\text{S} - 25^\circ\text{N}$, $89^\circ\text{W} - 15^\circ\text{E}$) because plans for ocean afforestation are not restricted to the *Sargassum* hotspots. Values for the entire map are mentioned just after those for the *Sargassum* hotspots (lines 158-160).

Re (iv): We generally agree with this statement. However, our calculation accounts for clouds as it is based on MERRA-2 surface incoming shortwave flux data product including clouds (and not their “clear sky” data product as assumed by the Reviewer). The difference between the two data products is shown in the figure below. It shows that clear sky conditions (lower row) lead to much higher surface incoming shortwave flux. The maps illustrate the 2018 average in W/m^2 and the timeseries show the development over the year in the study area as indicated by the rectangle. We repeated the calculation on changes in radiative forcing due to increasing albedo (equation 6 in supplement 8) with surface incoming shortwave flux from the “clear sky” dataset (bottom right timeseries in the figure below). As noted by the Reviewer, using the “clear sky” data product would lead to a more pronounced effect of albedo changes on radiative forcing ranging from 220-2200 PJ/year (in contrast to 180-1800 PJ/year we derive in the calculation using surface incoming shortwave flux including clouds). Thus, our calculation does not overestimate the surface albedo effect as we account for clouds.

Comment 4) The comments regarding monitoring and accounting are interesting and useful, but offer a too pessimistic view on the possible economic incentives to deploy marine afforestation as CDR measure. Terrestrial afforestation has economic value and is considered in international carbon trading even though it would not pass the criteria on permanence and accountability rightly viewed problematic by the authors for marine afforestation. I think a more balanced discussion is required to make this a useful scientific contribution to the debate.

REPLY: We agree and changed this text in the revised version (lines 187-208). We now discuss that sequestration has value even if short, but likely increases the longer the carbon remains sequestered (for which we add new references). We still point out problems, but maintain a neutral language on whether or not the problems are easy to solve. We also noted that the timescales of seafloor sequestration are substantially longer than in some terrestrial CDR approaches such as terrestrial afforestation and soil carbon sequestration. Overall, these changes should underline our neutral stance and provide balanced arguments.

individual points:

Comment 5) 1.21-23 not correct: ocean afforestation is not generally seen as key component of the marine portfolio.

REPLY: We disagree with this assessment by the Reviewer and think that our statement is correct for the following reasons.

- 1) Ocean afforestation is discussed as one of the main ocean-based CDR methods by the UN interagency working group on marine geoengineering (GESAMP, 2019).
- 2) The National Academy of Sciences (USA) is currently considering ocean afforestation as one of the 6 main methods in their upcoming report on ocean-based CDR (<https://www.nationalacademies.org/our-work/a-research-strategy-for-ocean-carbon-dioxide-removal-and-sequestration>).

- 3) There are a large number of start-ups beginning the implementation of ocean afforestation. E.g. Southern Ocean Carbon (<https://southernoceancarbon.com/>), Climate Foundation (<https://www.climatefoundation.org/marine-permaculture-sales.html>), or Green Wave (<https://www.greenwave.org/>) to name just three (more examples can be provided). We know of very few initiatives in any of the other three major ocean-based CDR approaches (Iron fertilization, alkalinity enhancement, artificial upwelling), suggesting that ocean afforestation is widely considered and even initialised.

Comment 6) 1.32 ‘alter...when upscaled’ is not correct. First, the effects do not depend on scale, second, they don’t alter the efficacy, but they determine the efficacy

REPLY: Agreed. We changed “alters” to “determine” and deleted “when upscaled” (lines 37-38).

Comment 7) 1.90-92. the N:P ratio of consumed nutrients (22.5) is higher than that of ordinary phytoplankton (16). The draw-down of fixed N might thus be topped by nitrogen fixation, leading to a smaller effect of nutrient reallocation.

REPLY: The N:P ratio of phytoplankton in the (sub)tropical North Atlantic is also more in the range of 22.5 and not 16 (Martiny et al., 2013). See also Re(ii) in response to comment 3.

Comment 8) 1.126-129. Why is permanent DOC storage essential to finance ocean afforestation? There is also no permanent C storage in afforestation on land, but still does not seem to affect funding or even consideration by UNFCCC.

REPLY: Agreed. We removed the word “permanent” from this sentence (line 146). The issue of permanence is discussed in a more neutral tone in the revised version of the manuscript (see reply to comment 4).

Comment 9) 1.135/136. The equilibration times shown in Fig.3 are high mostly north of 30N, but in the area of the GASB marked in Fig.1 are generally smaller than 6 (color scale is not fully appropriate to resolve the western tropical Atlantic and Caribbean).

REPLY: Agreed. We changed accordingly (see R(iii) in response to comment 3).

Comment 10) 1.163 A storage time of 700-900 years would be much longer than anything that can be achieved by terrestrial afforestation, which is considered in current UNFCCC carbon accounting.

REPLY: Agreed. We noted that this timescale is longer than some terrestrial methods (lines 199-201; see also our response to comment 4).

Comment 11) 1.215 Ref.40 is an extremely poor example of knowledge on the CDR potential of terrestrial afforestation. It should not be cited in this positive tone. The related comments have shown that ref.40 was poorly researched and did not present a fair understanding of the carbon cycle.

REPLY: Agreed. Our intention was to cite this study as one example where neglecting various feedbacks leads to misleading conclusions. This did obviously not come across so we modified the sentence structure and how we cite this study to make this clear in the revised version (lines 250-251).

Comment 12) 1.281 Chung reference is incomplete

REPLY: We thank the Reviewer for spotting this mistake and corrected this reference.

Comment 13) suppl. line 20 units of conversion factor missing.

REPLY: wet weight and TPC are both weights so the factor is g:g. We added this information (suppl. Line 40)

Comment 14) S3 1.94ff impact of N₂ fixation is neglected that could offset much/all? of the nitrogen reallocation.

REPLY: We show in Re(ii) as response to comment 3 that P limitation would lead to a similar reallocation effect. Thus, N₂-fixation would not change that because then P would become limiting with the same effect. Furthermore, we think that it cannot be assumed that N₂-fixation would top up an apparent N-deficit because N-fixation is iron- and not P-limited in the study region (Singh et al., 2017).

Comment 15) S8. Any change in sea surface albedo becomes effective for the planetary energy budget only in regions not covered by clouds. Changes in planetary albedo will in general be smaller than changes in surface albedo.

REPLY: See response Re (iv) to comment 3.

EVIEWERS' COMMENTS

Reviewer #2 (Remarks to the Author):

I apologize for the delayed review. I have now carefully read the response letter and the revised version and am satisfied with all of the authors responses and happy to recommend publication of this work.

Dear Editor,

All changes in the manuscript file are marked in yellow. No changes were made to the content. All changes correspond to moving the methods text that was previously in the SI to the “methods” section in the main manuscript file.

REVIEWERS' COMMENTS

Reviewer #2 (Remarks to the Author):

I apologize for the delayed review. I have now carefully read the response letter and the revised version and am satisfied with all of the authors responses and happy to recommend publication of this work.

Response to the Reviewer:

We thank the Reviewer for their time and constructive feedback that helped to improve this manuscript.

Kind regards,
Lennart Bach